# Development of Alkali Activated Inorganic Foams Based on Construction and Demolition Wastes for Thermal Insulation Applications

**DOI:** 10.3390/ma16114065

**Published:** 2023-05-30

**Authors:** Adrienn Boros, Gábor Erdei, Tamás Korim

**Affiliations:** Department of Materials Engineering, Faculty of Engineering, University of Pannonia, H-8201 Veszprém, Hungary; egabor97@gmail.com (G.E.); korim.tamas@mk.uni-pannon.hu (T.K.)

**Keywords:** construction and demolition waste, pozzolanic materials, waste concrete, alkali activated cement mortar, foam products, thermal insulating material

## Abstract

Nowadays, the construction industry is challenged not only by increasingly strict environmental regulations, but also by a shortage of raw materials and additives. It is critical to find new sources with which the circular economy and zero waste approach can be achieved. Promising candidates are alkali activated cements (AAC), which offer the potential to convert industrial wastes into higher added value products. The aim of the present research is to develop waste-based AAC foams with thermal insulation properties. During the experiments, pozzolanic materials (blast furnace slag, fly ash, and metakaolin) and waste concrete powder were used to produce first dense and then foamed structural materials. The effects of the concrete fractions, the relative proportions of each fraction, the liquid/solid ratio, and the amount of foaming agents on the physical properties were investigated. A correlation between macroscopic properties (strength, porosity, and thermal conductivity) and micro/macro structure was examined. It was found that concrete waste itself is suitable for the production of AACs, but when combined with other aluminosilicate source, the strength can be increased from 10 MPa up to 47 MPa. The thermal conductivity (0.049 W/mK) of the produced non-flammable foams is comparable to commercially available insulating materials.

## 1. Introduction

With the continuous growth of the population, as well as the development of industry and technology, the environment is being increasingly transformed; the size of untouched natural areas is decreasing. According to forecasts, more than 230 billion square meters of new buildings will be added to the built environment over the next 40 years. To put it in perspective, this is equivalent to adding Japan’s current floor area to the planet every year, the whole of New York City every 34 days, or Paris’s entire building stock every week [1,2]. Such large-scale construction projects require huge amounts of raw materials, and it is becoming increasingly difficult to provide the right quality and quantity of these materials. In addition, the fact that additives make up the bulk of a binder’s weight [3] is not negligible; nearly 70 wt% of aggregate is required for building construction, and 80–90 wt% of aggregate for road/highway construction. Based on indirect estimates, the world consumes roughly 40–50 billion tons of sand per year only for construction [4]. The growing demand for concrete is leading to serious legal/illegal extraction and, in some cases, a lack of natural (sand and gravel) sources [5]. By using recycled aggregates, it is possible to find an alternative way to handle this issue. For example, following careful collection and treatment, a significant part of construction and demolition wastes (e.g., waste concrete) could be used as concrete aggregates [6,7]. However, the lack of professional experience is still a major problem, so waste concrete is mainly used for road construction and backfilling [7].

The demand for cheaper and more widely available raw materials and a full life-cycle assessment has long been the primary motivation for sustainability and greener structures in the construction and building materials industry. In our dynamic world, the appearance of a new material often equals the absence of another material, so it is important that construction projects are carried out in the spirit of the green economy and a zero-waste approach. A significant part of the research is dealing with the possibilities of using alternative materials, mainly industrial wastes and by-products [6,7,8,9,10,11,12,13], since the amount of these materials is increasing year by year and of which, in many cases, the recovery is still an unsolved issue. The same is true for construction and demolition wastes (e.g., waste concrete, roof tiles, and bricks), whose production worldwide exceeds 3.5 billion tons per year [13]. It is a well-known fact that today the construction industry consumes 40% of raw materials and produces 35% of waste [7].

Alkali-activated cements (AAC), as prominent representatives of waste utilization, are becoming increasingly known material systems. One of the great advantages is that these systems can use wastes (e.g., construction and demolition wastes, rubber tires, glass, and mine tailings) not only as an aggregate [14,15,16,17], but also as a matrix [18,19,20,21,22,23]. Several industrial by-products and waste materials are recognized as raw materials [24]; e.g., in the case of blast furnace slag and fly ash, AACs can provide a solution for treating nearly 50 wt% of the waste generated [25]. AACs are not only able to utilize waste that would otherwise be landfilled, but at the end of their life cycle, they can also become an integral part of the circular economy model. The use of industrial waste as raw materials and aggregates for AACs not only contributes to sustainable construction, but also helps to protect the ecosystem. This idea is reinforced by the trend where, for example, the reuse of concrete materials demolished during road construction is increasingly coming into prominence. During this process, the concrete is crushed and separated into different fractions; the smallest size range (grain size < 4 mm) does not meet the requirements of the EN 12620 [26] standard and cannot be utilized as an aggregate. However, AACs can be produced from this waste material [27,28,29], among other reasons, because it contains most of the cementitious fraction.

The combination of industrial by-products and wastes from construction and demolition activities allows the production of higher value-added AACs with a solid fraction consisting exclusively of wastes. Much of the research is focused on traditional building applications [30,31,32,33,34], whereas foaming offers the possibility to produce lightweight elements with remarkable potential in higher added value applications (e.g., thermal/acoustic insulation, pH control, and air/wastewater treatment) [35,36,37,38,39,40,41,42,43,44]. Based on the literature, the use of construction and demolition wastes for the production of foamed AAC is not typical, probably because the resulting products have poor mechanical properties [27,28,29]. In foamed binders, such wastes are mostly used as aggregates without separation by type of material (concrete, bricks, glass, and wood) [45]. However, it may be beneficial to separate these wastes to ensure a constant level of quality.

The present study investigates “dense” and foamed AACs produced with the combination of concrete waste with more active raw materials (blast furnace slag, fly ash, and metakaolin). This paper extends to the exploration of the relationships between macroscopic properties (strength, porosity, and thermal conductivity) and micro/macro structure, the applicability of concrete waste as a raw material, as well as the potential to create larger foam building elements. This work can provide preliminary evidence that, by carefully choosing the experimental parameters, waste concrete can be used to produce higher value-added AACs that can contribute to the achievement of a green economy and zero-waste approaches. Based on the experiments performed in the scope of the present research, the foamed AAC produced from waste concrete can be deemed a promising thermal insulating material.

## 2. Materials and Methods

### 2.1. Materials

The raw material for the experiments was ground granulated blast furnace slag (GGBFS), which was provided by ISD DUNAFERR Zrt. (Dunaújváros, Hungary). For the purpose of comparison with other aluminosilicate sources, test specimens were prepared using Metaver R type metakaolin (MK) (Newchem GmbH, Baden, Austria) and fly ash (FA) (MVM Mátra Energia Zrt., Visonta, Hungary). The chemical composition of the starting materials (Table 1) was determined by X-ray fluorescence spectrometry (XRFS), the mineral composition (Table 2) by X-ray diffraction (XRD), and the amount of each phase by Rietveld Analysis.

Based on the XRD analysis of the raw materials (see Figure A1 in Appendix A), it can be said that all three materials contain a large amount of amorphous phase. In the case of GGBFS, the crystalline phases are merwinite, akermanite, bronwmillerite, and quartz, in the case of MK quartz, calcite, and hematite, and in the case of FA akermanite, quartz, hematite, and albite. The particle size distribution of aluminosilicate sources and their median (D50) were determined with the help of a laser granulometer, GGBFS: 25.8% < 5 μm, D50 = 13.84 μm; MK: 19.6% < 5 μm, D50 = 22.45 μm; FA: 7.9% < 5 μm, D50 = 42.62 μm (see Figure A2 in Appendix A).

For the preparation of AAC mortars, in the case of control samples, standard quartz sand (CEN Standard Sand according to DIN EN 196-1 [46], Normensand GmbH, Beckum, Germany) was used as a filler, while in the case of alternative aggregate mixtures, waste concrete (WC) powder from construction and demolition activities (Beton Technológia Centrum Kft., Budapest, Hungary) was used; the chemical composition of the latter is shown in Table 3. Based on the X-ray recording (see Figure A1 in Appendix A), the WC contains a small amount of amorphous phase, the main crystalline phase is quartz, and in addition, calcite, anorthite, larnite, and portlandite are also present as minor components (Table 4).

The maximum particle size of standard quartz sand is 2 mm, while that of waste concrete is 0.5 mm. The aggregates were divided into fractions with the help of sieve analysis, and the particle size distribution is shown in Table 5. The exact particle size distribution of waste concrete and the median (D50) were determined with the help of a laser granulometer: 5.5% < 5 μm, and D50 = 125.66 μm (see Figure A2 in Appendix A). The morphology of WC is shown in Figure A3 in Appendix A.

The alkaline medium (activating solution) required for the preparation of AACs was provided by a mixture of analytical grade NaOH pellets (Reanal Laborvegyszer Kereskedelmi Kft., Budapest, Hungary) and a commercially available sodium silicate solution (ANDA Kft., Barcs, Hungary). The chemical composition of the latter was as follows: 6.8 wt% Na_2_O, 28.6 wt% SiO_2_, and 64.6 wt% H_2_O.

During the foaming process, sodium oleate (Sigma-Aldrich Chemie GmbH, Steinheim, Germany) was used as a stabilizing agent, and a solution of 30 wt% of H_2_O_2_ (Reanal Laborvegyszer Kereskedelmi Kft., Budapest, Hungary) diluted to 4.5 wt% as a foaming agent (based on research settings by Boros et al. [35]).

### 2.2. Sample Preparation

In experimental work, “dense” and foamed AACs were prepared. In both cases, the activating solution was prepared in the same way: solid NaOH was dissolved directly in Na_2_SiO_3_ solution (and in distilled water). For the reason of reproducibility, all mixtures were prepared using an activating solution cooled to room temperature.

For the production of the AAC mortars (“dense”), the required amount of raw material (GGBFS, MK, or FA) was first measured, then the appropriately cooled activating solution was added. Identical conditions were ensured during the mixing, i.e., the slurry of raw material–alkaline solution was homogenized for 1 min at 900 rpm, then the aggregate (standard quartz sand or waste concrete) was added (with a 1:2 raw material: aggregate mass ratio). The resulting mortar was further mixed for 1 min at 900 rpm.

The steps for the preparation of AAC foams up to the addition of the foaming components were the same as for “dense” specimens (the slurry of the starting material–alkaline solution for 1 min at 900 rpm, +waste concrete for 1 min at 900 rpm). After homogenization of the waste concrete in the AAC slurry, sodium oleate was added. The resulting mixture was further stirred for 1 min at 1200 rpm, and finally, the H_2_O_2_ solution was added and homogenized for 1 min at 600 rpm.

The prepared fresh mixture was cast into ø30 mm × 30 mm cylindrical PVC molds for both “dense” and foamed AACs and was stored under ambient conditions (21–23 °C and RH = 50 ± 10%). Samples were demolded at 1 day of age, and the qualification tests were performed at 7 and 28 days of age. In this study, the 28-day average test results were reported.

In the first stage, GGBFS-based AAC mortar specimens were prepared using the following molar ratios: SiO_2_/Al_2_O_3_ = 6.6, Na_2_O/Al_2_O_3_ = 1.1, the mass ratio of Na_2_SiO_3_ to NaOH in the activating solution was 6.6, and that of the activating solution and blast furnace slag powder was 0.6. First, control samples containing standard quartz sand were prepared, then the total amount of the aggregate was replaced with waste concrete. Different liquid/solid (L/S) ratios (0.21, 0.315, 0.368, and 0.42) were used to determine the appropriate mixing consistency (Figure 1). The increase in liquid content of the mixtures was obtained by increasing the amount of activating solution, which modified the applied molar ratios as shown in Table 6.

The experimental work was continued by dividing the aggregates into various fractions, and then using the standard quartz sand and the waste concrete with the same particle size to prepare test specimens. The L/S ratio was varied between 0.315 and 0.42, as in the previous experimental phase.

To determine the reactivity of the construction and demolition waste used, AAC specimens were prepared using only waste concrete (without the addition of slag) by varying the L/S ratio (0.315–0.42). The molar ratios used were as follows: SiO_2_/Al_2_O_3_ = 3.4, 3.8, and 4.1, respectively, Na_2_O/Al_2_O_3_ = 1.4, 1.6, and 1.9, respectively. The composition of the alkaline medium was the same as the alkaline solution used in the previous sections, and the mass ratio of the activating solution to the concrete powder was 0.28, 0.32, and 0.37, respectively.

In the final phase of the mortar specimens test, AACs were produced by combining waste concrete with a particle size equal to the standard quartz sand and other more active raw materials (blast furnace slag, fly ash, and metakaolin). The SiO_2_/Al_2_O_3_ molar ratio in the fly ash-based mixture was 8.8, the Na_2_O/Al_2_O_3_ molar ratio was 1.9, the mass ratio of sodium silicate and sodium hydroxide in the activating solution was 14.3, and that of the activating solution and fly ash powder was 1.8. The following molar ratios were used in the metakaolin-based mixture: SiO_2_/Al_2_O_3_ = 7.6, Na_2_O/Al_2_O_3_ = 1.25, the mass ratio of Na_2_SiO_3_ to NaOH in the activating solution was 8.0, while the mass ratio of the activating solution to metakaolin powder was 1.2. The relative proportions of the components of the AAC samples prepared by combining the three different aluminosilicate sources and waste concrete are shown in Figure 2. The experimental parameters for each series of “dense” AACs are shown in Table 7.

In the second phase of the experimental work, foamed AAC specimens were prepared from the optimal mixtures of “dense” samples. The required amount of the foaming components (stabilizing agent and foaming agent) was determined in the case of GGBFS-waste concrete-based AAC. First, the amount of H_2_O_2_ solution (10 wt%) was fixed, and the stabilizer content (0, 0.1, 0.25, 0.5, and 1 wt% based on the weight of the AAC slurry (slag + concrete + activating solution)) was varied. The second time, taking into account the relevant physical properties, the sodium oleate content was fixed (0.1 wt%) and the amount of foaming agent was varied (15, 10, 8.75, 7.5, 6.75, and 5 wt%).

In the case of the foamed products, as with the “dense” specimens, the effect of the combination of more active raw materials and waste concrete on the physical properties was investigated. Accordingly, in parallel to the slag-based mixtures, fly ash and metakaolin-based foamed AAC specimens were produced using 0.1 wt% sodium-oleate and 7.5 wt% H_2_O_2_ solution.

Finally, the potential to create GGBFS—waste concrete-based foamed products on a larger size was investigated, with the green economy and zero-waste approach in mind. On the one hand, samples of dimensions 200 mm × 200 mm × 15 mm were prepared (for thermal conductivity measurement using the longitudinal heat flow meter method), and on the other hand, test specimens of dimensions 40 mm × 40 mm × 160 mm were prepared in accordance with the standard specifications used for cements (EN 196-1). The experiments for the foamed AACs, as in the case of the “dense” samples, consisted of several series, the details of which are shown in Table 8.

### 2.3. Methods

Qualitative and quantitative phase analyses of the aluminosilicate sources and waste concrete powder were carried out using a Philips PW 3710 type X-ray diffractometer with CuKα radiation (50 kV, 40 mA), graphite monochromator, and 0.02° 2Θ/s speed (in the 2Θ range, 10–70°). The X’Pert Data Collector software (version 2.2) was used to control the device and collect data. An internal standard method was used to determine the amounts of crystalline phases and the amorphous fraction. In the process, 0.1000 g of ZnO was added to 0.9000 g of powdered sample (maximum particle size < 63 μm). In order to avoid orientation, back-loaded samples were used in the tests. The evaluation of X-ray diffractograms and the implementation of Rietveld analysis were carried out with the help of X’Pert Highscore Plus and the ICDD PDF-2 reference database.

The chemical composition of the starting materials (GGBFS, FA, MK, and WC) was determined using a Philips Axios PW 4400/24 type wavelength-dispersive X-ray fluorescence analyzer with melt sample preparation (sample/solvent mass ratio = 1.8, solvent: Li_2_B_4_O_7_ + LiBO_2_) according to the EN 196-2:2013 standard.

The particle size distribution of GGBFS, FA, and MK and their median (D50) were determined using a Fritsch Laser Particle Sizer “Analysette 22”-type laser granulometer. The measuring range of the device is 0.1–1160 μm, the total measurement time is two minutes, and the red He-Ne laser beam used has a wavelength of 632.8 nm and a power of <3 mW. The required sample amount (0.5–1.0 g) was loaded manually, taking into account the appropriate feedback from the instrument (the optimum sample amount was indicated by the change of the scale color from red to green). In order to achieve appropriate dispersion and prevent intergranular aggregation prior to the start of the measurements, the samples were treated in a water bath equipped with an ultrasonic stirrer and pump for 60 s.

The compressive strength of all AAC samples and the flexural strength of the standard-size foamed specimens were determined with the help of a CONTROLS Automax5 automatic double chamber device with an upper measuring limit of 15/300 kN in terms of load. The measurements were performed according to the relevant cement standard (EN196-1), using a loading rate of 2400 N/s for compressive strength and 50 N/s for flexural strength tests. In the case of compressive strength tests, however, the sample sizes differed from the standard (ø30 mm × 30 mm cylindrical samples were also tested in addition to standard-size specimens). Prior to the start of the test, the surfaces of cylindrical specimens were filed down perpendicular to the side. With respect to each mixture, three parallel measurements were performed, and the average strength values were reported.

The bulk density and open porosity of the foamed AAC samples were determined using the Archimedes method. The specimens were placed in distilled water and boiled for 2 h. After boiling, the test specimens were cooled, and then their water-saturated weight was measured with the help of a hydrostatic balance in air and through immersion in distilled water. The density of the liquid medium was 1000 kg/m^3^. With respect to each mixture, three parallel measurements were performed, and the average bulk density and open porosity values with standard deviation were reported.

The thermal conductivity of the AAC foams was determined using a non-equilibrium method, namely, the modified transient plane source method (MTPS), according to the requirements of the ASTM D7984 standard. To carry out the measurements, C-Therm TCi equipment was used, with a measurement range of 0.002–220 W/mK. Prior to the start of the test, the surfaces of the specimens were filed down flat, and the dust in the exposed pores was removed. The thermal conductivity of the larger (200 mm × 200 mm × 15 mm) samples made from the optimal mixture was determined using an equilibrium method, namely, the longitudinal heat flow meter method (ASTM C518), with the help of a NETZSCH HFM 436/3/1 Lambda device (measurement range: 0.002–1 W/mK, 0–100 °C). Before starting the measurements, the samples’ surfaces were filed down perpendicular to the side.

Computed tomography (CT) images of the foamed specimens were made with the help of a Nikon XT H 225 ST type X-ray tomography and the associated VG Studio 3.4 software. During the measurements, a cathode current of 85 μA and an accelerating voltage of 160 kV were used (1250 projections per recording, 2 recordings/projection, and 500 ms data collection time).

## 3. Results and Discussion

### 3.1. The Production of AAC Mortar (“Dense”) Specimens

In the first stage of the experiments, GGBFS-based AAC mortar samples were prepared using standard quartz sand as a filler and waste concrete generated during construction and demolition activities.

#### 3.1.1. Experiments with Unfractionated Aggregates

As a first step, the control samples were prepared using standard quartz sand with a coarser particle size than waste concrete. Then, the total amount of inert aggregate was replaced by WC. While a liquid/solid ratio of 0.21 was sufficient for the workability of the sand-containing samples, this proved to be insufficient when using fine-grained waste concrete. In order to ensure the proper consistency of fresh mortar, mixtures with different L/S ratios were prepared. Figure 3 shows the compressive strength and bulk density values obtained (the letter S in the figure indicates the sand-containing samples, while the letter C indicates the samples containing concrete).

By replacing the aggregate, the relevant values of the control samples (with no waste concrete) (28.4 MPa and 2285 kg/m^3^) can be kept at the same level at the lowest F/S ratio (0.315). Although the further increase in the liquid/solid ratio facilitated the workability of concrete-containing mixtures, it did not always have a positive effect on the strength values. This was because the alkaline activation reaction, in contrast to the hydration experienced in the case of classic Portland cements, did not require a large amount of water, so that the excess water content (only necessary for workability) evaporated during the setting and solidification mechanisms, leaving pores, which led to a reduction in strength. Based on the results, it can be said that with the proper amount of activating solution (L/S = 0.368), the samples containing waste concrete can achieve higher strength than their counterparts produced using sand; a strength increase of ~13% was observed compared to the value of the control samples (from 28.4 to 32.2 MPa). The different results obtained using the two aggregates can be explained on the one hand by the different particle size distribution of sand and concrete (see Table 5) and the filler effect due to particle size, and on the other hand, by the fact that construction and demolition waste can serve as a raw material for AACs [27,29]. In the following, our goal was to prove these assumptions.

#### 3.1.2. Experiments with Fractionated Aggregates

Experiments were carried out to compare the relevant properties of specimens made using standard quartz sand and waste concrete of the same particle size distribution. As the quartz sands also contain fractions (1000–2000 and 500–1000 μm) that are not found in WC, only the fractions below 500 μm were used for further mixtures. Taking into account the finer fractions, the sand contains 42.5 wt% of particles between 250 and 500 μm, 57.0 wt% of particles between 63 and 250 μm, and 0.5 wt% of particles below 63 μm, and this particle size distribution was considered the relevant one for waste concrete. For both aggregates, three different liquid/solid ratios (0.315, 0.368, and 0.42) were used in order to investigate the behavior of the aggregates during the alkaline activation reaction. Figure 4a shows the strength and density results obtained, where S represents the sand-containing samples and C represents the concrete-containing samples (Figure 4b summarizes the results that facilitated the determination of the reactivity of the waste concrete powder).

For mixtures with the same activating solution content and particle size distribution, it was assumed that mixtures containing sand and concrete would have similar consistencies. However, contrary to expectations, the sand mixtures had lower viscosities than their counterparts containing waste concrete with the same activating solution content. For all three liquid/solid ratios, the waste concrete samples had the highest strength values. As in the previous stage of experiments, strength was maximized in this case using the 0.368 L/S ratio; a ~twofold strength increase compared to the value of the sand-containing samples (47.0 and 23.6 MPa, respectively) was observed. Since the sand and the waste concrete at this stage had the same particle size distribution, it can be concluded that the difference in the results obtained using the two aggregates was not due to the filler effect related to particle size.

#### 3.1.3. The Use of Waste Concrete as an AAC Raw Material

Based on the results and experience of the first and second series of experiments, it is likely that the waste concrete particles, unlike the sand particles, are involved in the alkali activation reaction. To demonstrate this, unfractionated WC was used to prepare AAC test specimens (without the addition of blast furnace slag). In this case also, three different L/S ratios were used to ensure the appropriate workability. Based on the results of Figure 4b, it can be concluded that the waste concrete is not an inert, non-reactive aggregate in the system but an active component. Hence, the waste concrete used in the present study is suitable as a raw material for AACs. In this test, also, the use of an F/S ratio of 0.368 was found to be the most favorable in terms of strength, but the strength obtained is barely above 10 MPa. Komnitsas et al. [27] made a similar finding in their research dealing with brick, tile, and concrete waste-based AACs. Their experiments have shown that the compressive strength values of the waste concrete-based mixtures were far below those of the other two construction and demolition waste-based AACs. The authors explained this by the different chemical composition of the raw materials and the formation/absence of chemical bonds during the alkali activation reaction. To increase the strength value of WC-based AACs, controlled conditions (controlled particle size distribution, determined activating solution composition (SiO_2_/Al_2_O_3_ and Na_2_O/SiO_2_ ratios), L/S ratio, and heat treatment [27,28,29]) and/or the addition of a more active raw material (e.g., blast furnace slag) are required. The experimental work was continued by investigating the latter parameter.

#### 3.1.4. The Use of Waste Concrete in Combination with More Active Raw Materials

In the final phase of the AAC mortar specimen test, the effect on strength and bulk density of the combination of waste concrete (with a defined particle size distribution) and other more active raw materials (blast furnace slag, fly ash, and metakaolin) was studied (Figure 5a).

For all three types of pozzolanic materials, the compressive strength values were significantly increased compared to the strength of samples containing only concrete waste (Figure 5b). The smallest improvement (~1.4-fold) is observed for fly ash, followed by the metakaolin-containing samples (~3.4-fold increase). The most significant strength result is obtained for mixtures containing ⅓ part blast furnace slag and ⅔ part waste concrete; in this case, an increase of 4.5-fold was achieved. Mahmoodi et al. [28] also investigated the effect of partial replacement of concrete waste with more active materials on the compressive strength. They found that a combination of waste concrete and blast furnace slag gave the best strength values. It was hypothesized that the type (N-A-S-(H) and/or C-(A)-S-H gel) and amount of reaction products formed during alkali activation of the aluminosilicate sources caused the difference in mechanical properties. The different results in the present study are due to the variation in the chemical composition of the starting components, including the reactive SiO_2_ + CaO content (see Table 9).

While the strength of the fly ash-based products (14.9 MPa) does not reach the value of the first strength class (32.5 MPa) according to the cement standard (EN 197-1), the relevant values of the other two more active material-based samples (36.1 and 47.0 MPa) are above it. Furthermore, using blast furnace slag, products can be developed with strengths exceeding the value of the second strength class (42.5 MPa). Therefore, it can be said that it has been proven possible to develop a waste-based AAC system whose physical properties are competitive with those of classical binders. Although all AACs were prepared using a 1:2 pozzolanic material:waste concrete mass ratio, it is important to note that there are differences in the waste content of the mixtures prepared using various aluminosilicate sources. Figure 2 shows that while the waste material content of mortars is below 50 wt% for systems containing metakaolin, mortars containing fly ash or blast furnace slag (the more active raw materials are also considered waste) are above 60 wt%. The AACs produced by combining WC with GGBFS contain the most waste (82.6 wt%). The resulting AACs, due to their physical properties, may be suitable for use as structural materials in places subject to static loading. Based on the literature, the foaming of AACs can be achieved under favorable conditions, and the porous products obtained may have promising properties [35,36,37,38,39,40,44]. With the production of lightweight/foamed AACs, not only conventional construction applications but higher value-added applications (e.g., thermal/acoustic insulation, pH control, and air/wastewater treatment) may also appear; thus, the present research work was continued in this spirit.

### 3.2. The Production of Foamed AAC Samples

In the second stage of the experiments, our goal was to produce special products, namely, foamed elements. In the development of the porous samples, efforts were made to fix the technological parameters (amount of foaming and stabilizing agent) that can be used to tailor the relevant properties (bulk density, porosity, and thermal conductivity) as desired. First, the blast furnace slag and waste concrete-based mixtures (using a 1:2 GGBFS:WC mass ratio) were foamed. It was specified that the product strength should reach or exceed 1 MPa, as this value corresponds to the minimum strength class for aerated concrete building elements, in accordance with the requirements of standard GB11968-2006. The purpose of the experiments was to produce foams with the lowest possible thermal conductivity, which can practically be considered a thermal insulating material (thermal conductivity < 0.065 W/mK according to the ISO and CEN standards). The success of this effort lies in the controllability of the pore system of foams, which, based on the relevant literature, can be best achieved by the combined technique of saponification/peroxide decomposition/gel casting [35,36,37,38]. In the process, fine metallic powders or hydrogen peroxide are most commonly used as blowing agents, but there are other alternatives. For example, Bhuyan et al. [39] were able to produce AAC foams with the help of peracetic acid. A combination of chemical foaming agents and surfactants is preferable to manage the relative proportion of closed and open pores, i.e., the macroporous network, as desired. In light of this, in the present study, foamed products were prepared by using a combination of sodium oleate and H_2_O_2_.

#### 3.2.1. The Effect of the Foam Stabilizer Amount on the Physical Properties

As a first step in the optimization of the foaming parameters, a fixed foaming agent concentration (4.5 wt%) and amount (10 wt% based on the weight of raw materials (GGBFS + WC)) were applied, while the content of sodium oleate was varied (0–1 wt% based on the weight of raw materials (GGBFS + WC)). Taking into account the relevant physical properties (compressive strength, thermal conductivity, bulk density, and apparent porosity; see Table 10 and Figure 6a), the foam stabilizer content was selected to achieve the most favorable results. Then, the amount of H_2_O_2_ (5–15 wt%) was varied at a fixed sodium oleate content to improve the physical properties of the products. The results obtained from the measurements are summarized in Figure 6b and Table 11.

With the increase of the amount of foam stabilizer, the apparent porosity of the samples increased, and the bulk density, compressive strength, and thermal conductivity values decreased accordingly. The samples containing 0.1 wt% sodium oleate were an exception, for which the compressive strength and thermal conductivity did not change in line with the open porosity. This can be explained by the mechanism of action of the saponification reaction in an alkaline medium. An increase in the amount of stabilizing agent increases the amount of surfactant formed in the strongly alkaline medium, which facilitates gas bubble stabilization [35,36,37,38]. It is important to note that this is only achieved if sufficient alkali content is available for both the alkaline activation and the saponification reactions. Otherwise, the excess foam precursor, in the present study sodium oleate, remained in its original form in the mixture and was not able to exert its beneficial effect. In our previous publication [35], we demonstrated that if the amount of surfactant formed in situ during the saponification reaction is not sufficient to stabilize the gas bubbles formed during H_2_O_2_ decomposition, the interconnected pore size in the cell walls increases, which reduces the load-bearing capacity of the products. In the present study, a foam stabilizer content of 0.1 wt% is the limit at which the foams produced were capable of meeting both of the mentioned requirements (compressive strength > 1 MPa, thermal conductivity < 0.065 W/mK); thus, this sodium oleate content was used for the further part of the experiments.

#### 3.2.2. The Effect of the Foaming Agent Amount on the Physical Properties

The amount of H_2_O_2_ solution was varied to improve the physical properties. As a first step, 5 and 15 wt% were chosen in addition to the existing 10 wt% foaming agent content. Then, in order to optimize the strength–conductivity relationship, intermediate measuring points were used: 8.25, 7.5, and 6.25 wt% H_2_O_2_ contents (see Figure 6b and Table 11).

Excessive amounts (15 wt%) of H_2_O_2_ visually increased the pore size, decreased the strength, and increased the porosity. The reduction of the amount of foaming agent had a positive effect on the pore structure obtained, resulting in smaller and more sphere-like pores, which led to an increase in the strength values. The foaming agent content should not be increased above 10 wt%, since the controllability of the physical properties (strength, porosity, and thermal conductivity) relevant for the application can be achieved with lower amounts. Based on the results, the mixture containing 7.5 wt% H_2_O_2_ is the most favorable choice for use as a thermal insulating material since it has the lowest thermal conductivity. The product, with a thermal conductivity of 0.049 W/mK, can be practically considered a thermal insulating material, which also has the proper strength (1.3 MPa) for the application. If the pore structure of specimens prepared with different amounts of stabilizing and foaming agents is compared (Figure 7), significant differences can be observed.

As the amount of foaming agent was increased, the pore size distribution of the foams became much more inhomogeneous, resulting in unfavorably large pores (average pore diameter: 3860 ± 840 μm) both in terms of compressive strength and thermal insulation properties (Figure 7a). The CT image (Figure 7b) of the product obtained using optimal foaming parameters shows that the pore size distribution is homogeneous, with only a small amount of larger pores (average pore diameter: 1210 ± 100 μm). Varying the amount of foam stabilizing agent has a less obvious effect on the average pore size (1320 ± 240 μm); only a few larger pores appear in the structure (Figure 7c), which is a result of the imperfections of the saponification reaction mentioned above.

#### 3.2.3. Blast Furnace Slag, Fly Ash, and Metakaolin-Based Foamed Products

Similar to the “dense” AACs, the macroscopic properties of the foamed products were investigated using waste concrete in combination with other more active raw materials (metakaolin and fly ash) instead of blast furnace slag (see Figure 2 for mixture compositions). The effect of varying the composition of the foams on the physical properties is summarized in Table 12 and Figure 8.

The results show that by combining waste concrete with more active raw materials, it is possible to produce lightweight/foamed elements with a thermal conductivity that meets the practical requirement for thermal insulating materials (thermal conductivity < 0.065 W/mK). While the strength of the products using slag and metakaolin met the 1 MPa requirement, the fly ash-based mixtures did not. In the latter case, a significant reduction in compressive strength was to be expected, as the fly ash-containing specimens also performed the worst among the “dense” AAC specimens. The pore structure of the AAC products prepared with the three different more active raw materials (Figure 9) consists of small, homogeneously distributed pores (average pore diameter is 1210 ± 100 μm in the case of slag, 1480 ± 240 μm for fly ash, and 1260 ± 130 μm for metakaolin-based AACs), but with fly ash and metakaolin (Figure 9b,c), a few larger pore can be found in the structure. The differences in macroscopic properties and micro/macro structures may be due to variations in the composition of the activating solutions and the quality of the raw materials.

Both slag and metakaolin-based mixtures can be used to produce a thermal insulating material. However, it is important to note that while slag is an industrial by-product, metakaolin is obtained by calcination, which significantly increases the cost of the raw material. Keeping this fact in mind, as well as green economy and zero-waste approaches, further experiments were carried out to determine the feasibility of using waste concrete-slag-based foamed products in larger sizes.

#### 3.2.4. Scale-Up Experiments

In order to find out whether the foaming method used in this study can be used to produce larger building elements, scale-up experiments were carried out. Considering that the mixture containing blast furnace slag achieved the most favorable results in the case of small samples (ø30 mm × 30 mm cylinder), this mixture was used for the last part of the experimental work. On the one hand, in accordance with standard specifications used for cements (EN 196-1), specimens of dimensions 40 mm × 40 mm × 160 mm were produced for flexural and compressive strength tests. On the other hand, a sample of dimensions 200 mm × 200 mm × 15 mm was prepared for thermal conductivity measurements using the longitudinal heat flow meter method. The results of the scale-up experiments are summarized in Table 13.

If the physical properties of the blast furnace slag and waste concrete-based porous AAC sample developed are compared with those of other commercially available thermal insulating materials, it can be concluded that the product investigated in this study may be suitable for reducing energy loss in buildings. Although the thermal conductivity of the produced waste-based AAC is higher than that of extruded polystyrene, foamed glass, mineral wool, expanded perlite, and cork board [40,41,42,43], it is similar to that of expanded clay and aerated concrete [42,43] and lower than that of wood fiber board [42] and gypsum plasterboard (ISO 10456). Furthermore, the tested waste-based AAC foam has a compressive strength that exceeds that of materials typically used for partition walls and external and internal insulation [44]. The waste concrete-based alkali-activated cement foams developed in this study have the potential to be used as thermal insulation materials in buildings. Due to its fire-resistant properties, it may be an excellent substitute for combustible polystyrene, and its malleability may give it an advantage over fibrous glass and rock wool insulation materials. It can be said that the material synthesized in this work (Figure 10) is non-flammable, rodent-resistant, can be easily cut/formed, has a simple production technology, and also contributes to environmental protection. Thus, it has remarkable potential for use in higher value-added applications.

## 4. Conclusions

This paper aimed to produce and test waste concrete-based AACs with remarkable potential for use in the scope of both traditional construction applications (“dense” specimens) and specific applications (lightweight/foamed elements). The primary goal of the research was to find the optimum for the experimental parameters that can be applied to develop AACs with controlled properties. The following conclusions can be drawn based on the experiments performed:Waste concrete from construction and demolition activities is not an inert, non-reactive aggregate in the alkali activation reaction but an active component. WC can be used as a raw material for AACs, but the strength of the product obtained (10.5 MPa) is significantly lower than that of conventional binders.To increase the strength value of waste concrete-based AACs, controlled conditions (controlled particle size distribution, L/S ratio) and the addition of a more active raw material (blast furnace slag, metakaolin, and fly ash) are required. The most significant strength improvement (from 10.5 up to 47.0 MPa) can be achieved with mixtures containing ⅓ part blast furnace slag and ⅔ part waste concrete.By combining waste concrete with various alumino-silicate sources, both AAC mortar specimens and foamed elements can be produced. The foaming process of waste concrete-based AACs can be carried out under favorable conditions, and the physical properties of the products (strength, porosity, and thermal conductivity) can be controlled by adjusting the amount of stabilizing and foaming agents.For use as a thermal insulating material, a mixture containing 0.1 wt% foam stabilizer and 7.5 wt% H_2_O_2_ is the most favorable choice; the thermal conductivity of the waste concrete-based product obtained in this way is 0.049 W/mK, and its strength is 1.3 MPa.Both waste concrete-slag and waste concrete-metakaolin-based mixtures may be suitable for the production of thermal insulating building elements.With the foaming method used in this study, the waste concrete-slag-based AAC can be produced in larger sizes. Considering the physical properties of the specimens of dimensions 40 mm × 40 mm × 160 mm and 200 mm × 200 mm × 15 mm, the developed foams have remarkable potential for use in thermal insulating applications.

On the whole, it can be concluded that, after a proper selection of the experimental parameters, waste concrete can be used to produce higher value-added AACs that contribute to the achievement of a green economy and zero-waste approaches. The physical properties of the developed waste concrete-based AACs are competitive with those of conventional binders in the case of “dense” elements and comparable with those of the most commonly used insulating materials in the case of foamed products.

## Figures and Tables

**Figure 1 materials-16-04065-f001:**
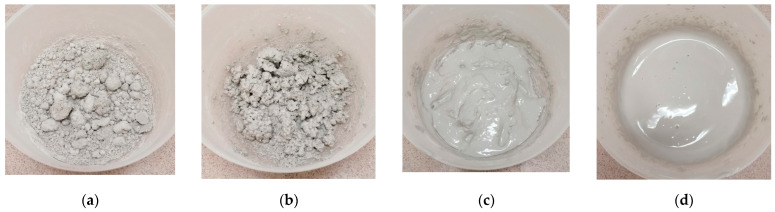
Consistency of mixtures prepared using different liquid/solid ratios ((**a**): 0.21, (**b**): 0.315, (**c**): 0.368, and (**d**): 0.42).

**Figure 2 materials-16-04065-f002:**
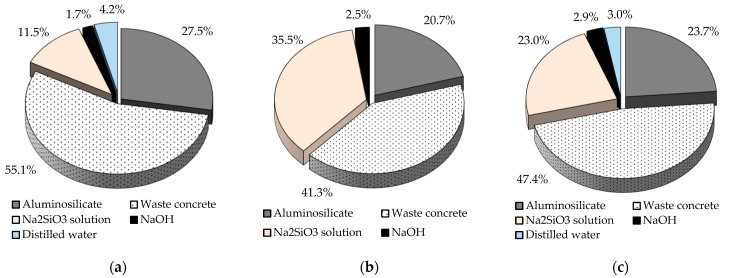
Compositions of AAC mortars obtained by combining waste concrete with (**a**) slag, (**b**) fly ash, and (**c**) metakaolin.

**Figure 3 materials-16-04065-f003:**
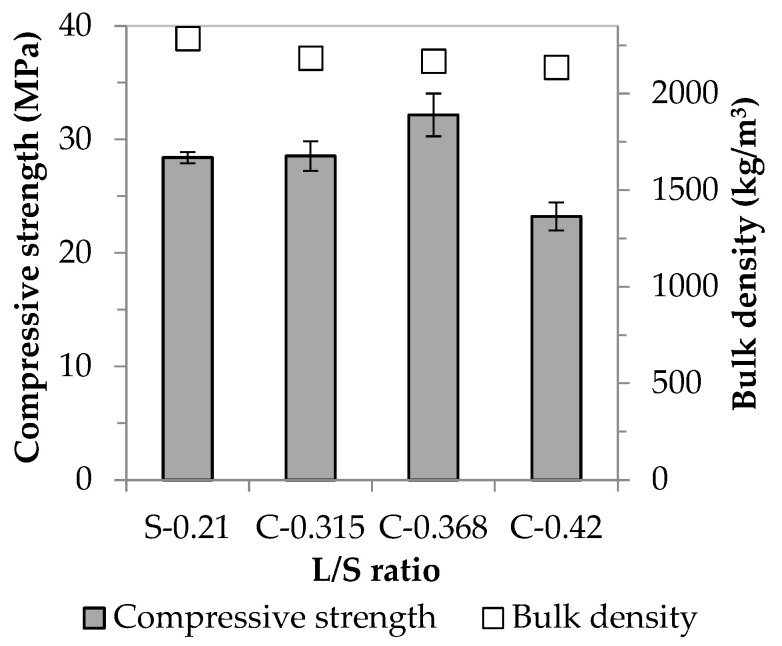
Physical properties of AAC specimens containing standard quartz sand (S) or waste concrete (C) produced using different liquid/solid ratios.

**Figure 4 materials-16-04065-f004:**
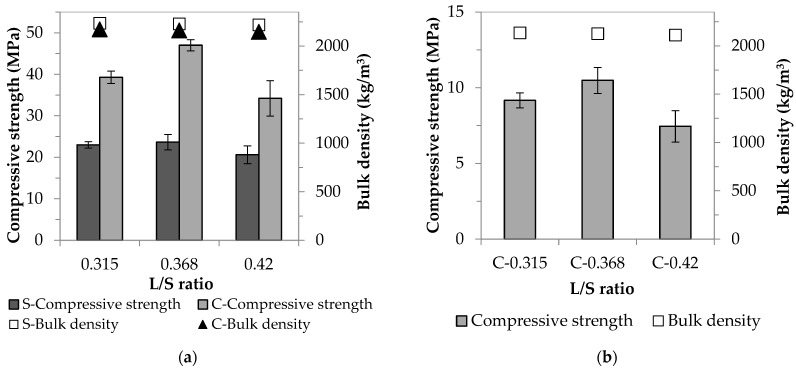
The effect of different L/S ratios on the compressive strength and bulk density of AAC specimens in the case of (**a**) aggregates with the same particle size distribution and (**b**) using only waste concrete as raw material.

**Figure 5 materials-16-04065-f005:**
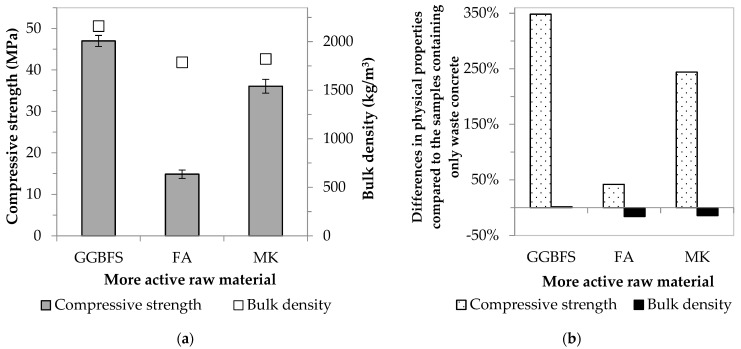
Effect of the combination of more active raw materials and waste concrete on (**a**) physical properties, and (**b**) the differences compared to a mixture based only on WC.

**Figure 6 materials-16-04065-f006:**
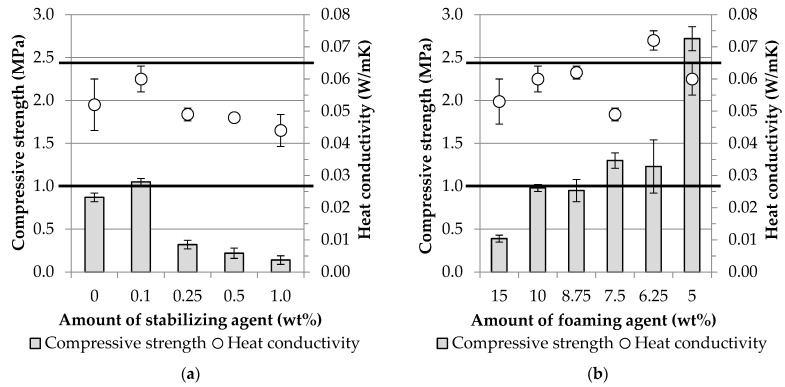
Effect of the amount of the foaming components ((**a**): sodium oleate and (**b**): H_2_O_2_ solution) on the compressive strength and thermal conductivity (Horizontal lines indicate boundary conditions in the case of the experimental job).

**Figure 7 materials-16-04065-f007:**
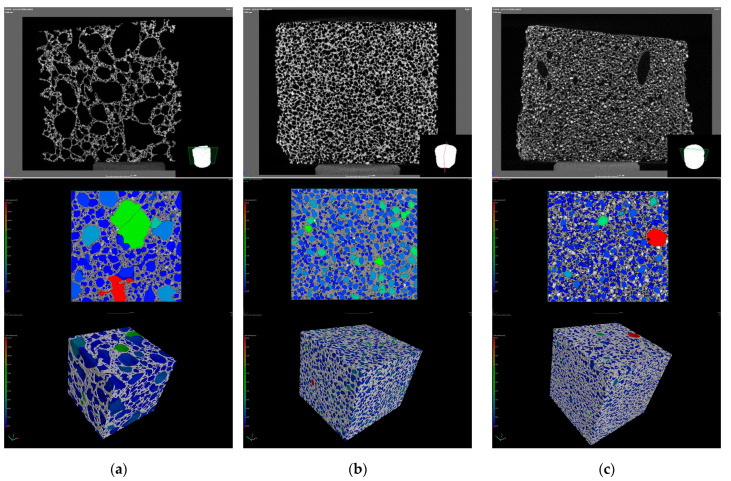
Effect of the amount of foaming and stabilizing agent on the pore structure ((**a**): 15/0.1, (**b**): 7.5/0.1, and (**c**): 10/1 H_2_O_2_ solution/sodium oleate content (wt%)).

**Figure 8 materials-16-04065-f008:**
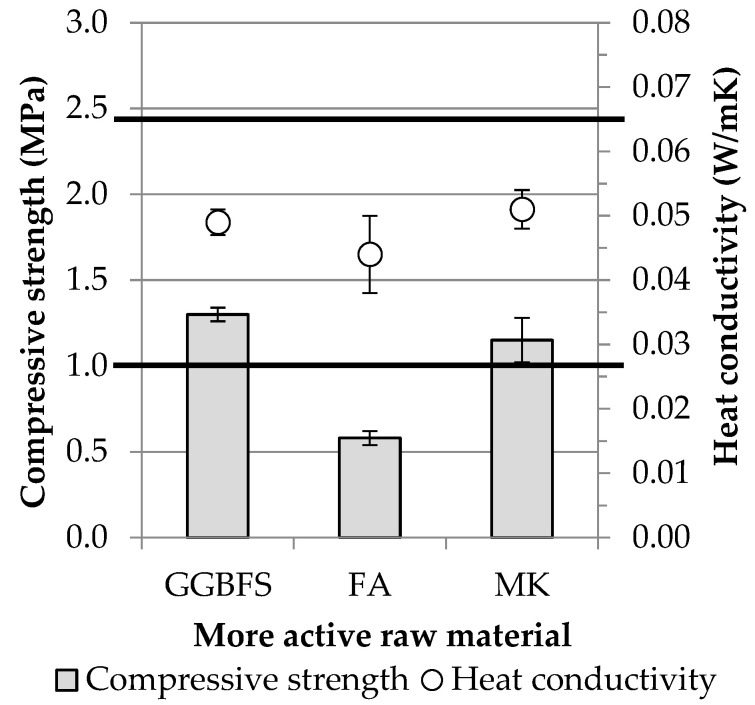
Effects on compressive strength and thermal conductivity of the combination of more active raw materials and concrete waste (Horizontal lines indicate boundary conditions in the case of the experimental job).

**Figure 9 materials-16-04065-f009:**
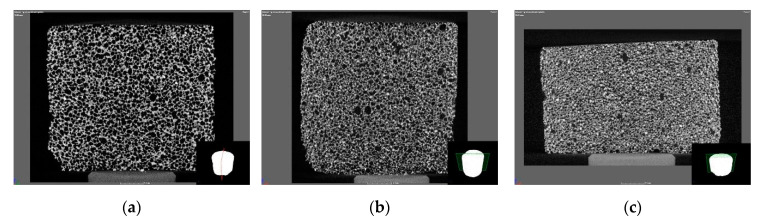
Effect of the combination of more active raw materials and waste concrete on the pore structure ((**a**): slag, (**b**): fly ash, and (**c**): metakaolin-based products).

**Figure 10 materials-16-04065-f010:**
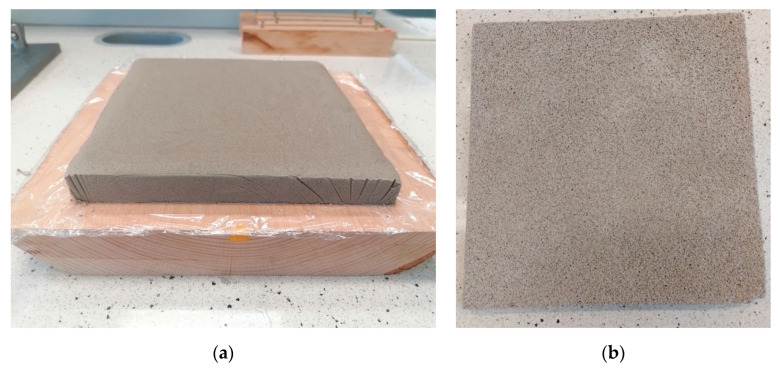
The blast furnace slag and waste concrete-based porous AAC product of dimensions 200 mm × 200 mm × 15 mm after demolding (**a**) and filed down perpendicular to the sides (**b**).

**Table 1 materials-16-04065-t001:** Chemical composition (wt%) of Slag (GGBFS), Metakaolin (MK), and Fly ash (FA).

	SiO_2_	Al_2_O_3_	CaO	MgO	Na_2_O	K_2_O	TiO_2_	Fe_2_O_3_	SO_3_	LOI
GGBFS	34.05	6.45	47.36	8.06	1.08	0.57	-	-	1.48	0.95
MK	66.58	21.09	2.08	0.49	0.05	0.43	1.03	4.37	0.03	3.85
FA	45.87	18.46	11.59	2.75	0.38	1.27	0.56	14.29	1.10	3.73

**Table 2 materials-16-04065-t002:** Phase composition (wt%) of Slag (GGBFS), Metakaolin (MK), and Fly ash (FA).

	Merwinite	Akermanite	Brownmillerite	Quartz	Calcite	Hematite	Albite	Vitreous Phase
GGBFS	11.1	1.2	0.6	0.2	-	-		86.9
MK	-	-	-	37.9	3.1	1.2	-	57.8
FA	-	4.5		7.2		4.3	6.7	77.3

**Table 3 materials-16-04065-t003:** Chemical composition (wt%) of Waste Concrete (WC).

	SiO_2_	Al_2_O_3_	CaO	MgO	Na_2_O	K_2_O	TiO_2_	Fe_2_O_3_	SO_3_	LOI
WC	71.88	4.00	12.05	0.65	0.61	0.69	0.17	1.46	0.39	8.71

**Table 4 materials-16-04065-t004:** Phase composition (wt%) of Waste Concrete (WC).

	Quartz	Calcite	Anorthite	Larnite	Portlandite	Vitreous Phase
WC	50.6	13.3	10.5	2.4	1.1	22.1

**Table 5 materials-16-04065-t005:** Particle size distribution of aggregates (wt%).

	Particle Size Range (μm)				
	1000–2000	500–1000	250–500	60–250	0–63
Standard quartz sand	37.5	30.9	13.4	18.0	0.2
Waste concrete	-	-	29.7	52.5	17.8

**Table 6 materials-16-04065-t006:** Experimental parameters for different liquid/solid ratios.

	Liquid/Solid Ratio			
	0.210	0.315	0.368	0.420
SiO_2_/Al_2_O_3_ molar ratio	6.6	7.5	7.9	8.4
Na_2_O/Al_2_O_3_ molar ration	1.1	1.7	2.0	2.3
Activating solution/slag mass ratio	0.6	0.9	1.1	1.2

**Table 7 materials-16-04065-t007:** Experimental parameters of “dense” AACs.

Series	Starting Material	Amount, wt%		L/S Ratio
		Standard Quartz Sand	Waste Concrete	
Experiments with unfractionated aggregates	GGBFS	100	0	0.210
		0	100	0.210–0.420
Experiments with fractionated aggregates	GGBFS	100	0	0.315–0.420
		0	100	0.315–0.420
The use of waste concrete as an AAC raw material	WC	0	100	0.315–0.420
The use of waste concrete in combination with more active raw materials	GGBFS	0	100	0.368
	FA	0	100	0.368
	MK	0	100	0.368

**Table 8 materials-16-04065-t008:** Experimental parameters of foamed AACs.

Series	Starting Material	Amount, wt%		Sample Size, mm
		H_2_O_2_	Sodium-Oleate	
Different foam stabilizer amount	GGBFS + WC	10	0–1	ø30 × 30
Different foaming agent amount	GGBFS + WC	5–15	0.1	ø30 × 30
The use of waste concrete in combination with more active raw materials	GGBFS + WC	7.5	0.1	ø30 × 30
	FA + WC	7.5	0.1	ø30 × 30
	MK + WC	7.5	0.1	ø30 × 30
Scale-up experiments	GGBFS + WC	7.5	0.1	40 × 40 × 160
				200 × 200 × 15

**Table 9 materials-16-04065-t009:** The reactive SiO_2_ and CaO content of the starting components (wt%).

	WC	FA	MK	GGBFS
Reactive SiO_2_ content	2.74	35.74	26.41	33.93
Reactive SiO_2_ content	8.51	11.12	0.87	47.03
Reactive SiO_2_ + CaO content	11.25	46.86	27.28	80.96

**Table 10 materials-16-04065-t010:** Effect of the foam stabilizer content on bulk density and apparent porosity.

	Amount of Sodium Oleate (wt%)				
	0.00	0.10	0.25	0.50	1.00
Bulk density (kg/m^3^)	778 ± 20	762 ± 19	646 ± 31	535 ± 17	508 ± 9
Open porosity (vol%)	43.8 ± 2.8	65.5 ± 3.2	68.4 ± 1.0	69.1 ± 1.4	76.8 ± 0.1

**Table 11 materials-16-04065-t011:** Effect of the foaming agent content on bulk density and apparent porosity.

	Amount of H_2_O_2_ (wt%)					
	15.00	10.00	8.25	7.50	6.25	5.00
Bulk density (kg/m^3^)	779 ± 7	762 ± 19	772 ± 4	658 ± 41	787 ± 11	957 ± 24
Open porosity (vol%)	74.6 ± 3.4	65.5 ± 3.2	64.2 ± 0.6	69.5 ± 1.8	64.0 ± 0.5	58.9 ± 1.5

**Table 12 materials-16-04065-t012:** Effect of the combination of waste concrete and more active raw materials on bulk density and apparent porosity.

	More Active Raw Material		
	Slag	Fly Ash	Metakaolin
Bulk density (kg/m^3^)	687 ± 41	740 ± 36	739 ± 12
Open porosity (vol%)	69.5 ± 1.8	63.9 ± 1.4	66.5 ± 0.7

**Table 13 materials-16-04065-t013:** Effect of scale-up on physical properties.

Flexural Strength (MPa)	Compressive Strength (MPa)	Bulk Density (kg/m^3^)	Open Porosity (vol%)	Heat Conductivity (W/mK)
0.4 ± 0.0	0.9 ± 0.1	703 ± 48	67.8 ± 1.7	0.113 ± 0.004

## Data Availability

Data are contained within the article.

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
