# Peer review of "Development of Alkali Activated Inorganic Foams Based on Construction and Demolition Wastes for Thermal Insulation Applications"

_materials, 2023, doi:10.3390/ma16114065_

Round 1

Reviewer 1 Report

This study aims to develop waste-based AAC foams with thermal insulation properties. During the experiments, pozzolanic materials (blast furnace slag, fly ash, metakaolin) and waste concrete powder were used to produce first dense and then foamed structural materials. The results showed concrete waste itself is suitable for the production of AACs, but when combined with other aluminosilicate source, the strength 20 can be increased from 10 MPa up to 47 MPa. 

The article is well research and contains novel idea that adds some information to the body of knowledge. Likewise, the paper complies with the writing standard of the Journal and all tests were done according to the normal standard of tests. Based on these aforementioned, I recommend that the research paper can be accepted for publication after minor revision.    

1.Please add “and” respectively between the fly ash and metakaolin in line 15 and between the porosity and thermal conductivity in line 18.  

2.Please capitalize the first letter of each keyword and modify it.

3.When multiple words or phrases are juxtaposed, please add a comma and an 'and' before the last word or phrase. For example, lines 17, 73, 97, and 124, as well as the entire text.

4.This article mainly studies the application of waste concrete as a raw material in alkali-activated materials, but the author only introduced it in the third paragraph of the introduction. Please modify it.                                    

5. What is the strength grade of the waste concrete used as the raw material in this article? How does the author handle obtaining the necessary raw materials? What are the microscopic morphological characteristics of the sample? Please supplement.

6.The author discussed on line 111 that waste concrete (WC) contains a small amount of amorphous phase. How can the author distinguish it from waste concrete? In fact, it is usually difficult to distinguish the diffraction peaks of XRD from years of abandoned concrete.

7. The particle distribution curve has an important influence on the performance of alkali-activated mortar and alkali-activated foam mortar. Please add.

8.Please remove “th” in line 149.

9. The mix proportion of blast furnace slag, fly ash, metakaolin, waste concrete, and standard sand required by the author in all tests is not clearly stated, so the author is requested to supplement it in the form of a three line table.

10. The author found through experiments that waste concrete is not an inert non-reactive aggregate in rows 314 and 316. May I know the activity index of this aggregate?

11.The theme of this article is the application of waste concrete in alkali-activated materials and alkali-activated foam materials. Therefore, the main conclusion needs to revolve around this theme. Please modify the conclusion.

This study aims to develop waste-based AAC foams with thermal insulation properties. During the experiments, pozzolanic materials (blast furnace slag, fly ash, metakaolin) and waste concrete powder were used to produce first dense and then foamed structural materials. The results showed concrete waste itself is suitable for the production of AACs, but when combined with other aluminosilicate source, the strength 20 can be increased from 10 MPa up to 47 MPa. 

The article is well research and contains novel idea that adds some information to the body of knowledge. Likewise, the paper complies with the writing standard of the Journal and all tests were done according to the normal standard of tests. Based on these aforementioned, I recommend that the research paper can be accepted for publication after minor revision.    

1.Please add “and” respectively between the fly ash and metakaolin in line 15 and between the porosity and thermal conductivity in line 18.  

2.Please capitalize the first letter of each keyword and modify it.

3.When multiple words or phrases are juxtaposed, please add a comma and an 'and' before the last word or phrase. For example, lines 17, 73, 97, and 124, as well as the entire text.

4.This article mainly studies the application of waste concrete as a raw material in alkali-activated materials, but the author only introduced it in the third paragraph of the introduction. Please modify it.                                    

5. What is the strength grade of the waste concrete used as the raw material in this article? How does the author handle obtaining the necessary raw materials? What are the microscopic morphological characteristics of the sample? Please supplement.

6.The author discussed on line 111 that waste concrete (WC) contains a small amount of amorphous phase. How can the author distinguish it from waste concrete? In fact, it is usually difficult to distinguish the diffraction peaks of XRD from years of abandoned concrete.

7. The particle distribution curve has an important influence on the performance of alkali-activated mortar and alkali-activated foam mortar. Please add.

8.Please remove “th” in line 149.

9. The mix proportion of blast furnace slag, fly ash, metakaolin, waste concrete, and standard sand required by the author in all tests is not clearly stated, so the author is requested to supplement it in the form of a three line table.

10. The author found through experiments that waste concrete is not an inert non-reactive aggregate in rows 314 and 316. May I know the activity index of this aggregate?

11.The theme of this article is the application of waste concrete in alkali-activated materials and alkali-activated foam materials. Therefore, the main conclusion needs to revolve around this theme. Please modify the conclusion.

Author Response

We thank the reviewer for their constructive comments and suggestions. We have revised our paper accordingly and feel that the reviewer’s comments helped clarify and improve our paper. I would like to answer your questions and comments point by point.

  • Comment 1: Please add “and” respectively between the fly ash and metakaolin in line 15 and between the porosity and thermal conductivity in line 18.

Response: Thank you for pointing this out. We agree with this comment; I have added "and" to the sentences in question.

  • Comment 2: Please capitalize the first letter of each keyword and modify it.

Response: Thank you for this suggestion. Some journals actually recommend the use of capital initials. However, Materials does not require this. The keywords were formatted according to the suggestions of the template used to prepare our manuscript. The first letter of each keyword has been lower-cased to meet the specifications. We hope that the Keywords part will be acceptable to the reviewer in its current state.

  • Comment 3: When multiple words or phrases are juxtaposed, please add a comma and an 'and' before the last word or phrase. For example, lines 17, 73, 97, and 124, as well as the entire text.

Response: We are really sorry that we have made this kind of mistake in several sentences. Thank you for pointing this out. We have subjected our manuscript to an extensive revision. During this process, where necessary, we have added a comma and an "and" to the sentences. We hope this will make our paper easier to read. We have marked all revisions to the manuscript using the "Track Changes" function in MS Word, such that any changes can be easily viewed by the editors and reviewers.

  • Comment 4: This article mainly studies the application of waste concrete as a raw material in alkali-activated materials, but the author only introduced it in the third paragraph of the introduction. Please modify it.

Response: The comment is justified; the applicability of concrete waste should be explained in the earlier paragraphs of the Introduction part. Accordingly, we have expanded the first two paragraphs of the manuscript. We hope that the Introduction part will be acceptable in its present state. However, if the reviewer feels that further modification is needed in relation to this section, please let us know and we will try to improve it.

  • Comment 5: What is the strength grade of the waste concrete used as the raw material in this article? How does the author handle obtaining the necessary raw materials? What are the microscopic morphological characteristics of the sample? Please supplement.

Response: The waste concrete used in this manuscript was collected from the demolition of the high-rise building in Pécs (also known as the 25-story building). It was once the tallest block house in Hungary, at 84 meters. The reinforced concrete structure of the building was constructed using IMS Yugoslav post-tensioning technology. For the construction of this type of building, normal architectural concrete with a strength of 30–35 MPa was normally used.

In the future, the sources for the raw materials needed could be the various building and road demolition operations. In my opinion, construction and demolition wastes as raw materials for AACs will be available in the long term due to the demolition and renovation needs of our time. Thus, an industry based on waste materials can remain viable.

The SEM micrographs of the waste concrete are available but were not included in the manuscript due to space savings. We admit that microscopic morphological characteristics can be important for the qualification of the raw material. In view of this, a SEM image of waste concrete has been added to Appendix A of the manuscript.

  • Comment 6: The author discussed on line 111 that waste concrete (WC) contains a small amount of amorphous phase. How can the author distinguish it from waste concrete? In fact, it is usually difficult to distinguish the diffraction peaks of XRD from years of abandoned concrete.

Response: For each raw material, the determination of the amorphous fraction and the identification of the crystalline phases were performed using X'Pert Highscore Plus software and the ICDD PDF-2 reference database. We agree that it is usually difficult to distinguish the diffraction peaks of XRD from years of abandoned concrete. In this case, in addition to software assistance, relevant literature references were used to identify the individual phases. To quantify the crystalline phases and the amorphous fraction, an internal standard method was used, whereby 0.1000 g of ZnO was added to 0.9000 g of powdered sample (maximum grain size <63 μm). Once the spectra were recorded, knowing the exact amount of internal standard (ZnO), the amount of each phase could be determined by Rietveld analysis with high accuracy. For the sake of better comprehensibility, we have supplemented the Materials and Methods part of the manuscript with the explanation given here.

  • Comment 7: The particle distribution curve has an important influence on the performance of alkali-activated mortar and alkali-activated foam mortar. Please add.

Response: The comment is justified; in addition to aluminosilicate sources, the particle size distribution of waste concrete can also have an important influence on the performance of "dense" and foamed alkali activated cements. In light of this, the particle distribution curve of waste concrete has been added to Appendix A of the manuscript.

  • Comment 8: Please remove “th” in line 149.

Response: I have removed “th” in line 149. Thank you for pointing this out.

  • Comment 9: The mix proportion of blast furnace slag, fly ash, metakaolin, waste concrete, and standard sand required by the author in all tests is not clearly stated, so the author is requested to supplement it in the form of a three line table.

Response: We regret that the reviewer feels that the mix proportion of blast furnace slag, fly ash, metakaolin, waste concrete, and standard sand required by us in all tests has not been clearly stated in the manuscript. We have tried to illustrate this in Figure 2 (in 2.2. Sample Preparation section). For the sake of clarity, we have prepared two tables summarizing the experimental parameters of this study. Hopefully, with the addition made, our manuscript will be easier to follow and understand for the reader and leave no more doubt about the mix proportions.

  • Comment 10: The author found through experiments that waste concrete is not an inert non-reactive aggregate in rows 314 and 316. May I know the activity index of this aggregate?

Response: I'm not sure I understand exactly what the reviewer means here. Would you be so kind as to clarify what activity index you mean? I highly appreciate your help.

In case the reviewer is interested in the pozzolanic activity, we do not have information on this at the moment. For waste concrete, the pozzolanic activity can be given, but the determination of this property is a time-consuming process, taking up to 15 days depending on the quality of the sample. However, if the reviewer feels the need, we will, of course, carry out the relevant experiments.

  • Comment 11: The theme of this article is the application of waste concrete in alkali-activated materials and alkali-activated foam materials. Therefore, the main conclusion needs to revolve around this theme. Please modify the conclusion.

Response: We agree with the reviewer's comment that the conclusions should reflect the most significant findings from the study. We have, accordingly, modified the Conclusion part to emphasize the application possibilities of waste concrete in "dense" and foamed alkali activated cements.

Reviewer 2 Report

The authors developed waste-based alkali-activated cement with thermal insulation properties in this study. The topic of this study is practically very significant, but certain shortcomings in the work should be eliminated before its consideration for publication.

1. Abstract should reflect the most significant findings from the study.

2. In the Introduction part, highlight the study's scientific contribution and summarize this work's methodology.

3. In part 2.3. provide additional information about used devices.

4. Compare the obtained results with available literature.

5.  General comment, application perspectives? Is the process economically viable?

Minor English corrections are needed.

Author Response

Thank you for the positive review and constructive comments. I would like to answer the questions and comments point by point.

  • Comment 1: Abstract should reflect the most significant findings from the study.

Response: We agree with the reviewer's comment. However, it is important to note that according to the requirements of the journal Materials, the Abstract should be a single paragraph of about 200 words maximum. Furthermore, the abstract should be an objective representation of the article and should not exaggerate the main conclusions. Taking these aspects into account, we have made an effort to give a pertinent overview of the work in the abstract. Based on the suggestions of the template used to prepare the manuscript, we have tried to write a structured abstract in the following style: background, aim, experimental parameters, results, and conclusions. Due to the word limit, we have not been able to include any more data in this section beyond the specific ones given. Therefore, the main findings of the study are presented in the Conclusion part. However, if the reviewer feels that a modification to the Abstract is needed, please let us know, and we will, of course, try to change it.

  • Comment 2: In the Introduction part, highlight the study's scientific contribution and summarize this work's methodology.

Response: We have revised and supplemented the Introduction part. We have tried to better highlight the novelty of the present research. We think that the methodology of the work is presented in sufficient detail in the further sections of the manuscript. For this reason and to avoid self-repetition, it has not been summarized in the Introduction part.

We hope that the Introduction part will be acceptable in its present state. However, if the reviewer feels that it is necessary to expand this section, please let us know, and we will try to supplement it.

  • Comment 3: In part 2.3. provide additional information about used devices.

Response: The comment is justified; more detailed descriptions of certain methods are needed to ensure the reproducibility of measurements. We have added additional information to section 2.3 of the manuscript.

  • Comment 4: Compare the obtained results with available literature.

Response: Thank you for pointing this out. We agree with this comment. We tried to discuss our results in this study in the light of the previously published relevant works. Taking into account the reviewer's comment, we have supplemented our manuscript where possible.

  • Comment 5: General comment, application perspectives? Is the process economically viable?

Response: The waste concrete-based alkali-activated cement foams developed in this study have the potential to be used as thermal insulation materials in buildings. Due to its fire-resistant properties, it may be an excellent substitute for combustible polystyrene, and its malleability may give it an advantage over fibrous glass and rock wool insulation materials. We have supplemented the manuscript with the general comments and application perspectives given here.

We think that the process is economically viable. Due to increasingly stringent environmental regulations and sustainable development-induced pressures to increase efficiency, the construction industry faces new challenges. The provision of raw materials and aggregates in the right quality and quantity is a daily challenge for professionals. In addition, the low ratio of waste management and recycling is also a serious problem. Waste is a sign of inefficiency, so we need to increasingly adapt our production processes to the green economy and zero-waste approaches. Instead of landfilling, alternative uses must be found that allow the production of higher value-added products. Particular emphasis should be placed on the management of construction and demolition waste. In fact, each road or building demolition process generates thousands of tons of concrete waste, only a very small proportion of which is recycled. An example is the demolition of the high-rise building in Pécs. Known also as the 25-story building, it was once Hungary's tallest block house, at 84 meters. The reinforced concrete structure of the building was constructed using IMS Yugoslav post-tensioning technology. In Hungary, similar techniques were very popular, but their immature application caused a number of problems, which ultimately resulted in the buildings becoming life-threatening. During the demolition of the high-rise building in Pécs, 22549 tons of concrete wastes were generated. According to available data, of this amount, around 5000-6000 tons were recycled, mostly for land filling, road construction, and parking lots. However, a significant proportion of the remaining waste is still awaiting alternative uses. The waste concrete-based alkali activated cement described in this study presented one possible alternative.

Reviewer 3 Report

In the present study, the authors investigated alkali activated inorganic foams based on construction and demolition wastes for possible usage in thermal insulation applications. They targeted to find out new solutions within the scope of circular economy model. In the design of the study, the authors evaluated pozzolonic materials and waste concrete powder. They focused on the impact of concrete fraction, liquid-to-solid ratio, and amount of foaming agent. Furthermore, some characterization techniques paved the way for understanding the effect of parameters. In my opinion, the thermal conductivity value of the prepared samples seem quite promising for thermal insulation applications. Therefore, I believe that this work has a good scientific sound and can be evaluated as a possible publication in this esteemed journal. Nevertheless, the authors should revise the following points, accordingly.

1-     The sample codes should be given in the abstract part so that the readers can comprehend the insights of the present investigation.

2-     The introduction part handles the main concept of the importance of the waste management. Yet more, the details of the alkali activation processes was successfully mentioned. Nonetheless, I certainly believe that the below-given literature studies have a great potential to support or even to extend the introduction part. I recommend authors to have a look and to make use of these studies, accordingly.

·        Bhuyan, M. A. H., Kurtulus, C., Heponiemi, A., & Luukkonen, T. (2023). Peracetic acid as a novel blowing agent in the direct foaming of alkali-activated materials. Applied Clay Science, 231, 106727.

·        Kinnunen, P., Ismailov, A., Solismaa, S., Sreenivasan, H., Räisänen, M. L., Levänen, E., & Illikainen, M. (2018). Recycling mine tailings in chemically bonded ceramics–a review. Journal of cleaner production, 174, 634-649.

3-     Table 2 should be revised in such a way that phase names can be written in a vertical position. It may be better to read out.

4-     In Archimedes’ density measurement, please ensure the density of the liquid medium used. Further, it will be better to see the tolerance values for this measurement.

5-     The results and discussions part were thoroughly given, and they seem pretty successful to reveal the main findings. However, I may recommend authors to compare their thermal conductivity results with the available literature studies, or even, with the commercially available ones. This option will certainly support the scientific soundness of the work.

Author Response

We appreciate the reviewer's helpful criticism and recommendations. After making the necessary revisions, we believe that the reviewer's suggestions helped to make our work clearer and better. I'd like to address each of your concerns and issues specifically.

  • Comment 1: The sample codes should be given in the abstract part so that the readers can comprehend the insights of the present investigation.

Response: I'm not sure I understand exactly what the reviewer means here. Would you be so kind as to clarify what sample codes you mean? I highly appreciate your help. Samples of the mixtures tested in this study were not individually labeled. The abbreviations used for the raw materials are those accepted in the literature: ground granulated blast furnace slag - GGBFS, fly ash - FA, metakaolin - MK, waste concrete - WC. These notations have not been included in the Abstract in order not to exceed the 200-word limit given by the journal. However, if the reviewer feels the need, we will, of course, insert these abbreviations in the Abstract.

  • Comment 2: The introduction part handles the main concept of the importance of the waste management. Yet more, the details of the alkali activation processes was successfully mentioned. Nonetheless, I certainly believe that the below-given literature studies have a great potential to support or even to extend the introduction part. I recommend authors to have a look and to make use of these studies, accordingly.
  • Bhuyan, M. A. H., Kurtulus, C., Heponiemi, A., & Luukkonen, T. (2023). Peracetic acid as a novel blowing agent in the direct foaming of alkali-activated materials. Applied Clay Science, 231, 106727.
  • Kinnunen, P., Ismailov, A., Solismaa, S., Sreenivasan, H., Räisänen, M. L., Levänen, E., & Illikainen, M. (2018). Recycling mine tailings in chemically bonded ceramics–a review. Journal of cleaner production, 174, 634-649.

Response: Thank you for this suggestion. The proposed articles contain valuable, useful information; it would be interesting to explore these methods. The use of peracetic acid as a foaming agent results in a low volume increase, which favors the use of the resulting foamed product for construction applications (e.g., sound and thermal insulation elements). Mining wastes can be valuable raw materials for alkali activated cements. Both recommended studies are good examples for the production of higher value-added products, and as such, they are a link to our manuscript and thus have been incorporated into it.

  • Comment 3: Table 2 should be revised in such a way that phase names can be written in a vertical position. It may be better to read out.

Response: Thank you for pointing this out. We partially agree with this comment. Perhaps if the names of the crystalline phases were written in a vertical position, it would help the reading. However, all tables in the manuscript are structured in a horizontal position because of the consistent structure. Furthermore, the vertical solution would take up a lot of space, so it might be preferable to keep the table in question in its original horizontal form. However, if the reviewer feels that it is necessary to rearrange the tables, we will, of course, make the modifications.

  • Comment 4: In Archimedes’ density measurement, please ensure the density of the liquid medium used. Further, it will be better to see the tolerance values for this measurement.

Response: For the determination of density by the Archimedes method, distilled water was used as the liquid medium, which has a density of 1000 kg/m3. We have added this information to the relevant part of section 2.3.

With respect to each mixture, three parallel measurements were performed, and the average bulk density values with standard deviation were reported. For the results given in the tables, the standard deviation values are given next to the bulk density data. For the results shown in the graphs, the standard deviation is added with an error marker. All diagrams include standard deviation values, but their absolute values are so small that those are hidden by the markers (typically in this case because of their size). The standard deviation of the bulk density of the mortar specimens is less than 30 kg/m3.

  • Comment 5: The results and discussions part were thoroughly given, and they seem pretty successful to reveal the main findings. However, I may recommend authors to compare their thermal conductivity results with the available literature studies, or even, with the commercially available ones. This option will certainly support the scientific soundness of the work.

Response: We have tried to provide information about this in section 3.2.4 of the manuscript (see scale-up experiments). For the small specimens (ø30 × 30 mm cylinders), we did not compare the obtained thermal conductivity with literature values. The reason is that large samples with dimensions of 200 mm × 200 mm × 15 mm are suitable for measurement under the standard conditions required by ASTM C518. The results obtained from these experiments may be suitable for predicting the applicability of the product in the construction industry. The thermal conductivity of the developed waste concrete-slag-based foamed alkali activated cement was compared with the relevant values of the following materials: extruded polystyrene, foamed glass, mineral wool, expanded perlite, cork board, expanded clay, aerated concrete, wood fiber board, and gypsum plasterboard. We have tried to give a comprehensive overview of the thermal insulation performance of commercially available thermal insulation materials and the product developed in our study. If the reviewer feels that something has been left out of the list, please let us know and we will add it.

Reviewer 4 Report

Questions to Authors of the manuscript "Development of alkali activated inorganic foams based on construction and demolition wastes for thermal insulation applications"

1. You indicated in line 472 that MK is expensive and increases the costs of the product, so why did you select MK for your study?

2. Line 477 - you indicated that your product can be used to produce larger building elements. However, you tested only 4x4x16 centimeter prisms to evaluate the strength of the matrix.  Dimensions are very close to those of cylindrical samples of 3,5cm in diameter you have already tested. So, 1) where is the difference? 2) What do you mean saying "larger building elements?"

3. 527 line - you mentioned "costs" in conclusions, but they are missing in your results part, so provide costs calculations data, please.

4. 531 line - you concluded that "specimens of  dimensions 40 mm × 40 mm × 160 mm and 200 mm × 200 mm × 15 mm, the developed foams have remarkable potential for use in thermal insulating applications", however, you tested the plate samples only for the conductivity. What can you say about other physical parameters of these plates? 

My suggestion is to revise the conclusion section.

Author Response

Thank you for the positive review and constructive comments. I would like to answer the questions and comments point by point.

  • Comment 1: You indicated in line 472 that MK is expensive and increases the costs of the product, so why did you select MK for your study?

Response: The reviewer's question is justified. I would like to briefly summarize the importance of metakaolin. The exact description of alkali-activated cements (AAC) is attributed to Davidovits, who studied the structure of the metakaolin-based systems. His work provided the basis for later research. To this day, metakaolin is one of the most commonly used solid precursors for the production of AACs; at least 50% of the relevant literature focuses on metakaolin-based systems. In this study, metakaolin obtained from heat treatment of kaolin was used to produce AACs. Calcination is necessary because kaolin is non-reactive in its original form and therefore cannot be used as a starting component for AACs. The heat treatment is an energy- and time-consuming process that entails costs but cannot be avoided. The kaolin needed to produce metakaolin is available in large quantities and is widely used in various industries (e.g., as raw material for porcelain production and paper filler). It is a pure raw material that is free from impurities. It is easy to work with, and there is no need to worry that the components it contains will take the product's properties in an undesirable direction. Thus, metakaolin can be considered a model material. For this and the reasons mentioned above, MK was chosen as one of the possible more active starting materials for our study.

  • Comment 2: Line 477 - you indicated that your product can be used to produce larger building elements. However, you tested only 4x4x16 centimeter prisms to evaluate the strength of the matrix. Dimensions are very close to those of cylindrical samples of 3,5cm in diameter you have already tested. So, 1) where is the difference? 2) What do you mean saying "larger building elements?"

Response: First of all, I would like to apologize. I misspelled the size of the small test specimens. Thank you for drawing my attention to this error. For the tests, ø30 mm × 30 mm cylinders were used instead of cylindrical samples of 35 mm in diameter. So the correct version is ø30 mm × 30 mm; I have corrected this in the manuscript.

It is important to note that in this study, we worked at a laboratory scale. At this level, the size of 40 mm × 40 mm × 160 mm is a significant improvement compared to ø30 × 30 mm. In the case of a cylinder, the cross-sectional surface area is 707 mm2, while in the case of a square column, it is 1600 mm2. In our opinion, the more than twofold increase in size is good for basic research. The results obtained in our study can serve as a basis for further research. Of course, a further aim is to produce a prototype of even larger size. If we look at the technological readiness levels (TRL 1–9), we can say that the present research is around levels 4-5. Our aim is to continue the project and reach levels 6–7 (large scale prototype and prototype system).

When we talk about the larger building elements in the manuscript, we are referring to the specimens of dimensions 40 mm × 40 mm × 160 mm and 200 mm × 200 mm × 15 mm produced in the study. The purpose of producing these was to demonstrate that foaming of waste concrete-based AACs can be achieved not only on a small size (laboratory scale) but also on a quasi-arbitrary size and shape. The larger dimensions (40 mm × 40 mm × 160 mm and 200 mm × 200 mm × 15 mm) we have investigated meet the requirements of the standard measurement specifications (EN 196-1 for compressive strength and ASTM C518 for thermal conductivity tests). For this reason, these dimensions were chosen.

  • Comment 3: 527 line - you mentioned "costs" in conclusions, but they are missing in your results part, so provide costs calculations data, please.

Response: Thank you for this suggestion. It would have been interesting to explore this aspect. However, in the case of our study, it seems slightly out of scope. The statement in the manuscript is factual, and we have not proved it with concrete data. We simply would have liked to point out that the two raw materials (slag and metakaolin) do not have the same costs due to their origin. Blast furnace slag is an industrial by-product generated in any case during the production of iron and is sufficiently reactive in itself. However, metakaolin is only available if naturally occurring kaolin is calcined. The latter process is highly energy-intensive, which increases costs. So while metakaolin is produced from a mined raw material, slag is considered a waste product. In this approach, it can be said without any calculation that slag-based systems are cheaper than their metakaolin counterparts. The cost analysis of the developed systems would require extensive research and is therefore not included in this study. However, if the reviewer feels the need, we will of course try to do a more detailed calculation, but we would ask for some time.

  • Comment 4: 531 line - you concluded that "specimens of dimensions 40 mm × 40 mm × 160 mm and 200 mm × 200 mm × 15 mm, the developed foams have remarkable potential for use in thermal insulating applications", however, you tested the plate samples only for the conductivity. What can you say about other physical parameters of these plates?

Response: The question is justified; for the 200 mm × 200 mm × 15 mm specimen, only the thermal conductivity was determined. This is the size at which thermal conductivity can be measured using the longitudinal heat flow meter method in accordance with ASTM C518. However, this size was not suitable for testing the compressive strength with the CONTROLS Automax5 device we used (the plate sample was too large to fit on the pressure head). Also, due to its size, we could not place it on the lower measuring point of the hydrostatic balance. The physical properties (strength, bulk density, and open porosity) of the larger specimens were determined using 40 mm × 40 mm × 160 mm specimens.

Furthermore, it is important to note that in this study, the composition of the waste concrete-slag-based AAC foams was the same for the ø30 mm × 30 mm cylinders and the larger specimens. Thus, the cylinders can be considered model material systems of testable size, which are also suitable for predicting the physical properties of larger specimens.

  • Comment 5: My suggestion is to revise the conclusion section.

Response: Thank you for this suggestion. We acknowledge that in some places we have used sloppy wording. We have revised the conclusions section and tried to improve it. As no cost calculation data was provided in the results section, we have removed this part from the conclusions. We hope that the Conclusion part will be acceptable to the reviewer in its current state.